# Smoking Awareness and Intention to Quit Smoking in Smoking Female Workers: Secondary Data Analysis

**DOI:** 10.3390/ijerph19052841

**Published:** 2022-03-01

**Authors:** Eun-Hye Lee, Sun-Hwa Shin, Goo-Churl Jeong

**Affiliations:** 1Nursing Department, College of Nursing, Sahmyook University, Seoul 01795, Korea; leeeh@syu.ac.kr (E.-H.L.); shinsh@syu.ac.kr (S.-H.S.); 2Department of Counseling Psychology, Sahmyook University, Seoul 01795, Korea

**Keywords:** female, smokers, smoking cessation, awareness, intention, smoke-free policy, decision making

## Abstract

Although the smoking rate among women has been continuously increasing recently, it is very difficult to explore the actual situation because of negative social views. This study aimed to analyze the effects of smoking awareness and living environment on the intention to quit smoking among female smokers. After receiving approval from the Research Ethics Committee in September 2021, secondary data analysis was performed for this study. A total of 378 working women who smoked were included in this study. The mean age was 34.4 years. The results showed that women living alone had significantly lower intentions to quit smoking, and women with experience in smoking cessation education had significantly higher intentions to quit smoking. In addition, it was found that the group having awareness of smoking cessation policy and smoking cessation treatment had high intention to quit smoking. As a result of the analysis of decision trees using data mining techniques, the strongest predictor of the intention of female workers who smoke to quit smoking was their perception of smoking cessation policies. In addition, it was found that the state’s policy support was important in that the group with the highest intention to quit smoking was the one with high awareness of both the smoking cessation policy and smoking cessation treatment. Finally, the risk group with the lowest intention to quit smoking was the group with low awareness of the anti-smoking policy, living alone, and having low awareness of the harmfulness of cigarettes. The importance of establishing policies for this vulnerable group, smoking cessation policies and treatment of female smokers, and improving awareness of the harmful effects of tobacco are discussed.

## 1. Introduction

Various regulations and policies on smoking helped decrease the smoking rate of adult men in Korea by 11.1%, from 47.8% in 2008 to 36.7% in 2018 [1]. However, the smoking rate for women underwent a relatively small change from 7.4% in 2008 to 7.5% in 2018, and the smoking rate has increased in women in their 20s to 40s [1]. The current trend of increasing smoking rates in women requires comprehensive examination. The National Health and Nutrition Examination Survey (NHNES) showed that the smoking rate in men decreased with age. In contrast, in women, the rate of smoking has increased in younger age groups [1]. The smoking rate of individuals in female 20s was 10.9%, indicating the need for interventions. In Korea, the smoking rate of men increased in the cohort born in the 1970s and decreased in the later cohort groups. However, for women, the smoking rate in cohorts born after the 1970s continues to increase [1]. A nationwide survey to develop a measurement tool for cigarette smoking reported a smoking rate of 11.1% among women [2]. According to the NHNES, the smoking rate is greater than 10% in women (10.9%), suggesting the need to seek active interventions for smoking women.

Recently, active participation of women in society has also led to an increased smoking rate [3]. However, there is a greater social stigma for smoking in women than in men, which causes women to hide their smoking behavior [4]. Additionally, such negative perceptions of female smokers lead to difficulties in quitting smoking and benefits from smoking cessation support services [5]. Female smokers showed different smoking behaviors than males. Women tend to smoke more cigarettes at once than men when given the opportunity, and they show a strong urge to smoke when alone [6]. Therefore, it is important to consider the environmental characteristics of the workplace for female office workers who smoke to identify the factors that induce smoking [7]. The workplace provides women with a reason to smoke and provides a smoking environment. Therefore, smoking cessation efforts for female workers must include individual efforts and improved organizational interventions that account for factors of the workplace environment [8]. 

The most effective factor for smoking cessation is a smoker’s voluntary intention to quit [9], and intention to quit smoking is a prerequisite for any preparation for and practice of smoking cessation [10]. Additionally, according to planned behavior theory, the decisive factor in smoking cessation is the intention to quit smoking [11]. Therefore, to implement effective smoking cessation policies, authorities must understand the intention to quit smoking among smokers [12]. In studies on intention to quit smoking conducted in Korea, older female adults were reported to have a lower intention to quit smoking than their male counterparts [13]. In another study of female smokers, the chance of intending to quit smoking was higher in those who made previous attempts to quit smoking, had a high health-related quality of life, and had experience with smoking cessation campaigns [14]. However, as the number of female smokers continues to rise with the increasing role of women in society, identifying the factors of the work environment that affect the intention to quit smoking among female smokers would be meaningful.

As women tend to hide their smoking behaviors from others [15], studies on smoking-related awareness in female smokers are limited. Smoking awareness has significant effects on smoking attitudes [16], and incorrect awareness and positive attitudes toward smoking increase the likelihood of smoking [17]. Therefore, understanding smoking awareness in women may help predict smoking behavior and evaluate their intention to quit smoking.

The decision tree analysis technique classifies and predicts subgroups according to the rules of decision-making and helps explain the relative effects of factors in specific states using a visual tree structure [18]. The decision tree analysis technique has been a useful method for exploring the factors of participant behavior using data related to participants [19,20]. This technique has also been used to identify the educational characteristics that affect adolescents’ intention to quit smoking [21]. Therefore, in this study, we assessed the relationship between smoking awareness and intention to quit smoking in female smokers and identified the factors that affected the intention to quit smoking using decision tree analysis. The findings of this study are expected to contribute to the formulation of policies that promote the intention to quit smoking among female smokers.

This study aimed to evaluate the relationship between smoking awareness and intention to quit smoking among female smokers. The specific aims of the study were as follows. First, to assess the occupational and smoking-related characteristics of female smokers. Second, there were differences in the intention to quit smoking according to the occupational and smoking-related characteristics of the female smokers. Third, we assessed the correlation between smoking awareness (awareness of harm, smoking cessation treatment, and smoking cessation policies) and intention to quit smoking. Fourth, we explored the factors affecting the intention to quit smoking among female smokers through a decision tree analysis.

## 2. Materials and Methods

### 2.1. Research Design

This descriptive study sought to assess the occupational and smoking-related characteristics of female workers who smoked and identify the factors that affected their intention to quit smoking. We conducted secondary data analysis using data from 378 smoking female workers from a study on the development of a cigarette- and smoking-related awareness measurement tool, conducted with the support of the Korea Health Promotion Institute in 2020. 

### 2.2. Study Participants and Data Collection

Primary data were collected from 1000 smokers nationwide (500 male smokers and 500 female smokers) from 4–8 December 2020. Among the 500 female smokers, data from 378 female workers were included in the secondary data analysis. In this study, 122 smokers without a job were excluded from the analysis. 

### 2.3. Instruments

#### 2.3.1. Intention to Quit Smoking

The intention to quit smoking was assessed using the International Tobacco Control Four Country Survey, which has been widely used in studies on smoking cessation [22]. The item “do you plan to quit smoking in the future” was used, and among the four possible responses, “within one month,” “within 6 months,” and “someday” were thought to indicate intention to quit smoking. The response “not at all” was considered to indicate having no intention of quitting smoking. The responses were analyzed as dichotomous variables.

#### 2.3.2. Smoking Awareness

Smoking awareness was evaluated using the instrument called tobacco and smoking-related awareness measurement tool by subject [2]. The preliminary items of the tool were derived from a review of previous studies, expert advice, open-ended surveys, and in-depth interviews, followed by expert appropriateness verification and Delphi analysis. The tool consisted of 39 items, including 11 on harmfulness awareness, 14 on smoking cessation treatment awareness, and 14 on smoking cessation policy awareness. The items were evaluated on a 5-point Likert scale, ranging from 1 (strongly disagree) to 5 (strongly agree). The higher the score for smoking harmfulness awareness, the more accurate the perception; the higher the score for smoking cessation treatment awareness, the more positive the awareness of physical medical services for smoking. The higher the score for smoking cessation policy awareness, the more positive was the awareness of the government’s smoking regulations and smoking cessation service support policies. To analyze the predictive factors affecting the intention to quit smoking using decision tree analysis, we divided harmfulness awareness, smoking cessation treatment awareness, and smoking cessation policy awareness into upper and lower groups based on the mean, which were analyzed as dichotomous variables. Cronbach’s α was 0.91 for harmfulness awareness, 0.84 for smoking cessation treatment awareness, and 0.82 for smoking cessation policy awareness [2]. In our study, Cronbach’s α was 0.80 for harmfulness awareness, 0.72 for smoking cessation treatment awareness, and 0.89 for smoking cessation policy awareness.

#### 2.3.3. Demographic, Occupational, and Smoking-Related Characteristics

Participants’ characteristics were divided into three areas based on previous studies [8,23,24,25,26]. The demographic characteristics included age, body mass index (BMI), education level, marital status, and cohabitation. Occupational characteristics included workplace size, title, employment type, number of smoking co-workers, physical labor intensity, emotional labor intensity, performance pressure, work hour autonomy, and job satisfaction. Smoking-related characteristics included desire to smoke (1–10 points), number of cigarettes smoked, smoking type, and experience of smoking cessation education. 

### 2.4. Ethical Consideration 

This study was approved by the institutional review board of Sahmyook University (IRB no. 2-1040781-A-N-012021120HR) on 1 September 2021. 

### 2.5. Data Analysis

IBM SPSS Statistics for Windows, Version 25.0 (IBM Corp., Armonk, NY, USA) was used for statistical analysis of the collected data. Frequency analysis was conducted on the demographic, occupational, and smoking-related characteristics of the participants. χ^2^-test was conducted to determine differences in the intention to quit smoking according to the characteristics of the participants. Pearson’s correlation coefficient analysis was conducted to determine the correlation between smoking awareness and intention to quit smoking. To identify the predictive factors affecting the intention to quit smoking, we performed decision tree analysis. We used the classification and regression tree algorithm to simplify the decision tree analysis model. A total of 21 predicting variables were included in the analysis. The model was segmented using a depth of five, a parent node with a minimum size of 30, and a child node with a minimum size of 15. The gains index was calculated to compare the smoking intention ratio of the entire group with the smoking intention ratio of each node. The group with a gain index ≥100% was interpreted as having a high intention to quit smoking. Return on investment (ROI) calculates profit by using revenue and expense for each category of the target variable. ROI was evaluated “no intention to quit smoking” as expense (profit = −1) and “intention to quit smoking” as revenue (profit = 1). To validate the decision tree analysis, we calculated the risk estimate and performed a k-fold evaluation (k = 10) for cross validation. 

## 3. Results

### 3.1. Demographic, Occupational, and Smoking-Related Characteristics

Table 1 presents the demographic characteristics. A total of 129 (34.1%), 163 (43.1%), and 86 (22.8%) participants were in their 20s, 30s, and 40s, respectively. Regarding BMI, 69 (18.3%) participants were under-weight, with a BMI score lower than 19; 258 participants (68.3%) were in the normal range of 20–24, and 51 participants (13.5%) were overweight, with a BMI score greater than 25. A total of 283 (74.9%) participants had graduated from university or higher, and 207 (54.8%) were unmarried. Additionally, 77 participants (20.4%) lived alone, whereas 137 (36.2%) and 164 (43.3%) lived with nonsmokers and smokers, respectively. Emotional labor intensity was strong in 211 participants (55.8%).

A total of 232 participants (61.4%) worked in small and medium-sized businesses, and most of the participants were employees (307 participants, 81.2%). Regarding employment type, 318 (81.4%) participants were regular workers, whereas 60 (15.9%) were non-regular workers. Moreover, 53 (14.0%), 111 (29.4%), 132 (34.9%), and 82 (21.7%) participants had 0, 1–2, 2–9, and >10 smoking co-workers, respectively. A total of 162 (42.6%) participants answered that their level of physical intensity at work was low, and 164 (43.3%) and 53 (14.0%) responded to normal and strong physical intensity at work, respectively. For the item on pressure for performance at work, the greatest number of participants (206 participants, 54.5%) answered that they were not pressured, followed by 90 (23.8%) and 82 (21.7%) participants who responded “neutral” and “pressured.” A total of 117 (31.0%) participants answered that they had autonomy for their work hours, whereas 183 (48.4%) participants responded “neutral” and 78 (20.6%) answered that they could freely adjust their work hours. Regarding job satisfaction, 60 (15.9%), 183 (48.4%), and 135 (35.7%) participants were unsatisfied, neutral, and satisfied, respectively.

The survey revealed the following smoking-related characteristics. A total of 207 (54.8%) participants had a strong desire to smoke, whereas 171 (45.2%) had a low desire to smoke. Regarding smoking amount per instance, 223 (59.0%), 144 (38.1%), and 11 (2.9%) participants smoked 1–2, 3–10, and >10 cigarettes, respectively. A total of 151 (39.9%) participants smoked out of “habit/dependence,” whereas 129 (31.1%) and 65 (17.2%) smoked for “stress relief” and “effect-seeking,” respectively. Additionally, 293 (77.5%) participants did not have experience in smoking cessation education. A total of 178 (47.1%), 164 (43.4%), and 172 (45.5%) participants had high awareness of the harmfulness of smoking, smoking cessation treatment, and smoking cessation policy, respectively.

### 3.2. Differences in Intention to Quit Smoking according to Demographic, Occupational, and Smoking-Related Characteristics

We conducted a χ^2^ test to assess the differences in the intention to quit smoking among smoking female workers according to their demographic characteristics (Table 1). Among the participants, 72.7% of those who lived alone intended to quit smoking. This was significantly lower than those who lived with non-smokers (82.5%) and smokers (86.0%) (χ^2^ = 2.27, *p* = 0.044). We found no significant differences in the intention to quit smoking according to occupational characteristics. Regarding smoking-related characteristics, the intention to quit smoking was 89.4% in those who had experience of smoking cessation education, which was significantly higher than in those without experience of smoking cessation education (79.9%) (χ^2^ = 4.07, *p* = 0.028). In terms of awareness of smoking cessation treatment, the number of participants with the intention to quit smoking was significantly higher in those with high awareness than in those with low awareness (88.4% vs. 77.1%; χ^2^ = 8.05, *p* = 0.005). Furthermore, the number of participants with the intention to quit smoking was significantly higher in the group with high awareness of smoking cessation policy than in those with low awareness (91.9% vs. 73.8%; χ^2^ = 20.76, *p* < 0.001).

### 3.3. Correlation between Smoking Awareness and Intention to Quit Smoking

Table 2 shows the mean and standard deviation of smoking awareness and the intention to quit smoking. Smoking cessation treatment awareness was positively correlated with intention to quit smoking (r = 0.17, *p* = 0.001), as was smoking cessation policy awareness (r = 0.38, *p* < 0.001). Awareness of the harmfulness of smoking was not significantly correlated with the intention to quit smoking.

### 3.4. Decision Tree for Intention to Quit Smoking 

The demographic, occupational, and smoking-related characteristic scales consisted of categorical variables that did not satisfy the assumption of normality. Thus, we applied the decision tree of the nonparametric analysis method. The target variable of the analysis was intention to quit smoking, and all predictor variables were categorical variables.

The analysis showed 13 nodes, seven terminal nodes, and three depths. The risk estimate, which indicates the risk of misclassification, was as low as 0.17 (SE = 0.02). Cross-validation showed that the risk estimate was 0.19 (SE = 0.02), and the two risk estimates yielded no significant differences, suggesting that the analysis results were adequate.

Figure 1 illustrates the structure of a decision tree. The majority (82.0%) of female smokers had the intention to quit smoking at the root node. The strongest predictor variable for classifying the intention to quit smoking was smoking cessation policy awareness (improvement = 0.016). A majority (91.9%) of those with high awareness of smoking cessation policies had the intention to quit smoking, which was significantly higher than that of those with low awareness (91.9% vs. 73.8%).

We also assessed the gain index of the terminal nodes (Table 3). The gain index was the greatest for terminal node 6 (116.3%) and terminal node 11 (116.0%). A total of 28.6% of the total smoking female workers were classified as terminal node 6, and 95.4% of those with a high awareness of smoking cessation policy and treatment had the intention to quit smoking. Meanwhile, 10.8% of the total smoking female workers were classified as terminal node 11, and 95.1% of those with high smoking cessation policy awareness, low smoking cessation treatment awareness, and low job satisfaction intended to quit smoking.

Although, in the case of nodes 9, 8, and 12, the gains index was <100%, the ROI value was >100%. Terminal node 9 was the largest terminal node, with 34.9% of the total smoking female workers, and 81.1% of the participants showed the intention to quit smoking. Participants in the node had low smoking cessation policy awareness, did not live alone, and were autonomous in terms of their work hours. These findings offer new insights into the working environment and the cohabitation of female smokers. In terminal node 10, which was composed of similar participants who had autonomy for their work hours, the ROI was as low as 70.7%. Exactly 63.0% had the intention to quit smoking, suggesting that the intention to quit smoking decreased with low awareness of smoking cessation policies in those who could freely adjust their work hours. Although only 6.6% (25 participants) were in terminal node 7, the ROI was negative (21.4%), and the proportion of those with no intention to quit smoking (56.0%) was higher than that of those with intention to quit smoking (44.0%), suggesting that these participants were at a high risk of smoking cessation intention. Those in terminal node 7 had low smoking cessation policy awareness, lived alone, and had low awareness of the harmfulness of smoking. Terminal node 8 had similar conditions as terminal node 7 but had a high awareness of smoking harmfulness. The intention to quit smoking increased to 77.3% in terminal node 8, highlighting the importance of awareness of the harmfulness of smoking (Table 3).

## 4. Discussion

This study assessed intention to quit smoking based on the awareness of female workers who smoked. A decision tree analysis was conducted to investigate the characteristics of smoking female workers who intended to quit smoking. The results identified important variables for the intention to quit smoking in female smokers and provided evidence for smoking cessation policies and interventions.

This study confirmed that smoking cessation policy awareness was the most important variable predicting the intention to quit smoking. In particular, intention to quit smoking was significantly higher in the group with high awareness of smoking cessation policy than in the group with low awareness (91.9% vs. 73.8%). Various countries have proposed policies that are suitable for each country’s characteristics, and cooperate with other countries to formulate effective smoking cessation policies [27,28]. In a previous study, adults exposed to a smoking cessation campaign had significantly higher smoking cessation intentions than those who did not [14], and in this study it was similar in that the smoking cessation intention was high when the awareness of the smoking cessation policy was high. Therefore, it is important to publicize the contents of the smoking cessation policy to raise awareness, as it is directly related to the promotion of smoking cessation intention. Effective smoking cessation policies contribute the most to increasing the intention to quit smoking among smokers, and reinforcement of smoking cessation policies provides the motivation to attempt to quit smoking, especially in women [29,30]. Therefore, more interest and effort are required to develop smoking cessation policies for female smokers. Publicizing the contents of smoking cessation policies to raise awareness is directly related to increasing the intention to quit smoking and is fundamentally important to reducing the smoking rate. Unlike male smokers, female smokers in Korea are exposed to negative social stigma [5]. The development and implementation of smoking cessation policies that consider such social stigma may provide a stronger motive to quit smoking among women who experience social prejudice. However, despite active smoking cessation policies, the number of women who attempt to quit smoking and continuously receive smoking cessation counselling remains significantly low [31]. To raise awareness of smoking cessation policies, effective strategies must be sought by considering the characteristics of female smokers, whose smoking rate has not been accurately assessed [15].

Smoking cessation treatment awareness acted as an important predictor variable when female workers’ smoking cessation policy awareness was high. This finding implies that understanding how female smokers are undergoing smoking cessation treatment, and how effective the treatment is, are essential steps in increasing the intention to quit smoking in women. As previously described, female smokers show a passive attitude toward visiting treatment centers for smoking cessation [31], and lack information on specific therapeutic approaches to quit smoking [29]. In addition, those who are reluctant to disclose their smoking behaviors to others experience difficulties in asking for help, and hesitate to participate in active treatment in their attempt to quit smoking [32]. Our data showed that 77.5% of women who smoked had never undergone smoking cessation education, and a low number had access to smoking cessation education, which is the first step in smoking cessation. Therefore, it is necessary to create a platform for information on customized smoking cessation treatment for female smokers, including hidden smokers who are not identified, to improve smoking cessation treatment awareness.

When female workers’ smoking cessation policy awareness was low, residence type was an important factor. In particular, those who lived alone and had low awareness of the harmfulness of smoking had a high risk of being in the risk group with the lowest intention to quit smoking (terminal node 7). A higher proportion of those who lived alone and had a high awareness of the harmfulness of smoking reported an intention to quit smoking (33.3%). A previous study reported that smokers had a higher intention to quit smoking when they lived with their spouses [13]; however, it was difficult to compare them directly with this study because they did not report the smoking status of their spouses. These results show that policies to raise awareness of smoking harmfulness are urgently needed for female smokers living alone. The number of single-person female households has continuously increased in Korea since the 2000s, and female single-person household members have a high level of health risk behaviors and lack health-promotion-related behaviors [33,34,35]. These findings are consistent with ours, thereby emphasizing the importance of research interest in the smoking behavior of female workers living alone, providing correct information on the harmfulness of smoking, and recommending active smoking cessation treatment. In particular, as the reduced harmfulness and double effects of e-cigarettes are highlighted through advertisements [36,37,38], many female smokers are increasingly using e-cigarettes to avoid the negative stigma of cigarette smoking [39]. This phenomenon may increase the smoking rate among women in the future. Female single-person households who smoke are a vulnerable group with relatively poor health behavior, lack the resources to cope with emergencies [40], and must receive continuous management for smoking cessation treatment. Following the rapid development of smartphones and advancement of the Internet, the dangers of smoking may be effectively relayed to female smokers [41,42]. Therefore, the intention to quit smoking may be increased by improving the level of health literacy for smoking prevalence in single-person households through educational interventions on the awareness of smoking harmfulness.

The intention to quit smoking was significantly lower in those who were autonomous in terms of their work hours (terminal node 10) and in those who were satisfied with their current occupation (terminal node 12). These findings suggest that occupational characteristics that guarantee work satisfaction and autonomy over work hours are related to intention to quit smoking. A previous study showed that perceived stress of adults had a positive effect on intention to quit smoking [14], and this study had a similar aspect in that the group dissatisfied with their job had a high intention to quit smoking. Additionally, these results imply that the burden of smoking and the need to quit smoking are lower in those who work as freelancers or have professional occupations, as they have fewer restrictions on smoking in their work environment. In previous studies, those with high job demands and low autonomy showed a high rate of smoking [43,44]. These findings showed that low job satisfaction and high job autonomy led to no changes in the smoking rate. In contrast, we observed that those with similar characteristics but low autonomy for work hours had a higher intention to quit smoking. As the smoking rate was high despite the high intention to quit smoking, such smokers may have difficulty starting or practicing smoking cessation owing to their occupational characteristics. Therefore, as the intention to quit smoking and practice of smoking cessation may differ, it would be necessary to provide opportunities to practice smoking cessation for those with low autonomy for work hours. Additionally, identifying the behaviors of occupational characteristics that induce smoking may provide basic data for developing direct interventions to practice smoking cessation, including an increased intention to quit smoking. Therefore, further studies must be conducted to identify the environmental factors in the workplace that are related to the intention to quit smoking among female smokers in Korea who face smoking-related social stigma. In a previous study on workplace smoking cessation programs, the establishment of a healthy working environment and health promotion programs during the break time of female workers in call-centers with a high smoking rate significantly reduced the smoking rate [45]. Therefore, based on the effects of workplace smoking cessation programs [45,46], desirable health promotion behaviors need to be promoted through active smoking cessation treatment programs for female workers who smoke.

A few limitations must be considered when interpreting the results of this study. As this was a secondary data analysis study, there were limitations in the selection and composition of variables for measuring their effects on the intention to quit smoking. Additionally, not all physical and psychological factors at the individual level that affect the intention to quit smoking could be fully reflected. Nonetheless, the significance of this study is its nationwide sampling of female workers who smoke, whose data were not readily accessible. In addition, we assessed smoking awareness and occupational factors, showing that further studies on smoking cessation policies for female workers who smoke are needed. Our study also emphasized the importance of actively providing smoking cessation treatment and programs for female workers who smoke and provided basic data for the development of relevant programs.

## 5. Conclusions

This study was conducted to identify the factors affecting the intention to quit smoking among Korean female workers who smoke. As a result of the study, the proportion of such workers who had an intention to quit smoking was 82.0%. The strongest predictor of intention to quit smoking was perception of smoking cessation policy, followed by perception of smoking cessation treatment. This suggests that the establishment and expansion of national policies are important, and efforts are needed to increase awareness of the national smoking cessation support center. In contrast, the high-risk group with a high percentage of ‘no intention to quit’ was the group with low awareness of smoking cessation policy, living alone, and low awareness of the harmful effects of tobacco. Therefore, applying an education program is required to increase awareness of the harmfulness of tobacco as the most urgent measure for risk groups with low intention to quit smoking. Considering the overall research results, it is important to increase awareness of effective smoking cessation policies and treatments to increase the intention to quit smoking among female workers.

## Figures and Tables

**Figure 1 ijerph-19-02841-f001:**
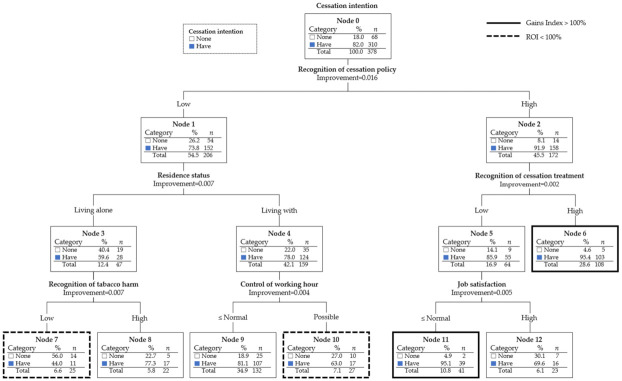
Prediction model of smoking cessation intention of smoking female worker by smoking awareness.

**Table 1 ijerph-19-02841-t001:** Differences in the Intention to Quit Smoking according to the General and Workplace Characteristics of Participants (*n* = 378).

Characteristics	Categories	*n* (%)	Intention to Quit Smoking		χ^2^ (*p*)
			No (*n* = 68)	Yes (*n* = 310)	
			*n* (%)	*n* (%)	
Age (year)	20–29	129 (34.1)	17 (13.2)	112 (86.8)	3.12 (0.210)
	30–39	163 (43.1)	34 (20.9)	129 (79.1)	
	≥40	86 (22.8)	17 (19.8)	69 (80.2)	
Body mass index	≤19	69 (18.3)	12 (17.4)	57 (82.6)	0.27 (0.875)
	20–24	258 (68.3)	48 (18.6)	210 (81.4)	
	≥25	51 (13.5)	8 (15.7)	43 (84.3)	
Education level	Highschool diploma or lower	95 (25.1)	22 (23.2)	73 (76.8)	2.30 (0.089)
	University degree or higher	283 (74.9)	46 (16.3)	237 (83.7)	
Marital status	Single	207 (54.8)	42 (20.3)	165 (79.7)	3.77 (0.151)
	Married	148 (39.2)	20 (13.5)	128 (86.5)	
	Other	23 (6.1)	6 (26.1)	17 (73.9)	
Cohabitant	Live alone	77 (20.4)	21 (27.3)	56 (72.7)	2.27 (0.044)
	Non-smoking cohabitant	137 (36.2)	24 (17.5)	113 (82.5)	
	Smoking cohabitant	164 (43.4)	23 (14.0)	141 (86.0)	
Workplace size	Large corporations/public institutions	62 (16.4)	10 (16.1)	52 (83.9)	0.45 (0.799)
	Small and medium businesses	232 (61.4)	41 (17.7)	191 (82.3)	
	Private businesses	84 (22.2)	17 (20.2)	67 (79.8)	
Workplace title	Employee	307 (81.2)	52 (16.9)	255 (83.1)	1.22 (0.174)
	Manager/management position	71 (18.8)	16 (22.5)	55 (77.5)	
Employment type	Regular	318 (84.1)	57 (17.9)	261 (82.1)	0.01 (0.532)
	Non-regular	60 (15.9)	11 (18.3)	49 (81.7)	
Number of smoking co-workers (number)	None	53 (14.0)	12 (22.6)	41 (77.4)	6.92 (0.074)
	1–2	111 (29.4)	26 (23.4)	85 (76.6)	
	3–9	132 (34.9)	22 (16.7)	110 (83.3)	
	≥ 10	82 (21.7)	8 (9.8)	74 (90.2)	
Physical labor intensity	Low intensity	161 (42.6)	34 (21.1)	127 (78.9)	1.94 (0.379)
	Neutral	164 (43.4)	25 (15.2)	139 (84.8)	
	High intensity	53 (14.0)	9 (17.0)	44 (83.0)	
Emotional labor intensity	Low intensity	28 (7.4)	7 (25.0)	21 (75.0)	1.41 (0.495)
	Neutral	139 (36.8)	22 (15.8)	117 (84.2)	
	High intensity	211 (55.8)	39 (18.5)	172 (81.5)	
Pressure for work performance	Not pressured	206 (54.5)	38 (18.4)	168 (81.6)	0.33 (0.847)
	Neutral	90 (23.8)	17 (18.9)	73 (81.1)	
	Pressured	82 (21.7)	13 (15.9)	69 (84.1)	
Work hour autonomy	Not autonomous	117 (31.0)	24 (20.5)	93 (79.5)	3.65 (0.161)
	Neutral	183 (48.4)	26 (14.2)	157 (85.8)	
	Autonomous	78 (20.6)	18 (23.1)	60 (76.9)	
Job satisfaction	Unsatisfied	60 (15.9)	8 (13.3)	52 (86.7)	4.77 (0.092)
	Neutral	183 (48.4)	28 (15.3)	155 (84.7)	
	Satisfied	135 (35.7)	32 (23.7)	103 (76.3)	
Desire to smoke	Low	171 (45.2)	37 (21.6)	134 (78.4)	2.82 (0.062)
	High	207 (54.8)	31 (15.0)	176 (85.0)	
Amount of smoking per instance (number)	1–2	223 (59.0)	40 (17.9)	183 (82.1)	2.69 (0.261)
	3–10	144 (38.1)	24 (16.7)	120 (83.3)	
	≥11	11 (2.9)	4 (36.4)	7 (63.6)	
Smoking type	Behavioral/dependent	151 (39.9)	33 (21.9)	118 (78.1)	4.99 (0.172)
	Stress relief	129 (34.1)	16 (12.4)	113 (87.6)	
	Stimulus-seeking/boredom relief	65 (17.2)	14 (21.5)	51 (78.5)	
	Effect-seeking	33 (8.7)	5 (15.2)	28 (84.8)	
Smoking cessation education	No	293 (77.5)	59 (20.1)	234 (79.9)	4.07 (0.028)
	Yes	85 (22.5)	9 (10.6)	76 (89.4)	
Smoking harmfulness awareness	Low	200 (52.9)	41 (20.5)	159 (79.5)	1.82 (0.183)
	High	178 (47.1)	27 (15.2)	151 (84.8)	
Smoking cessation treatment awareness	Low	214 (56.6)	49 (22.9)	165 (77.1)	8.05 (0.005)
	High	164 (43.4)	19 (11.6)	145 (88.4)	
Smoking cessation policy awareness	Low	206 (54.5)	54 (26.2)	152 (73.8)	20.76 (<0.001)
	High	172 (45.5)	14 (8.1)	158 (91.9)	

**Table 2 ijerph-19-02841-t002:** Descriptive Statistics and Correlation of Research Variables (*n* = 378).

Variables	Smoking Awareness	Smoking Harmfulness Awareness	Smoking Cessation Treatment Awareness	Smoking Cessation Policy Awareness	Mean ± SD	Skewness	Kurtosis
	r (*p*)						
Intention to quit smoking	0.32 (<0.001)	0.02 (0.735)	0.17 (0.001)	0.38 (<0.001)	2.36 ± 0.96	0.34	−0.81
Smoking awareness		0.40 (<0.001)	0.69 (<0.001)	0.85 (<0.001)	3.18 ± 0.37	0.64	0.95
Smoking harmfulness awareness			−0.12 (0.022)	−0.01 (0.873)	2.58 ± 0.59	0.10	0.80
Smoking cessation treatment awareness				0.53 (<0.001)	3.58 ± 0.41	−0.01	0.89
Smoking cessation policy awareness					3.24 ± 0.66	0.27	−0.23

**Table 3 ijerph-19-02841-t003:** Gains Index and ROI for Nodes (*n* = 378).

Node	Node		Gain		Response	Gains Index	Profit	ROI
	n	Percentage	n	Percentage				
6	108	28.6%	103	33.2%	95.4%	116.3%	0.907	1960.0%
11	41	10.8%	39	12.6%	95.1%	116.0%	0.902	1850.0%
9	132	34.9%	107	34.5%	81.1%	98.8%	0.621	328.0%
8	22	5.8%	17	5.5%	77.3%	94.2%	0.545	240.0%
12	23	6.1%	16	5.2%	69.6%	84.8%	0.391	128.6%
10	27	7.1%	17	5.5%	63.0%	76.8%	0.259	70.0%
7	25	6.6%	11	3.5%	44.0%	53.7%	−0.120	−21.4%
ROI = Return on investment.								

## Data Availability

The data presented in this study are available upon request from the authors.

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
