# Peer review of "Smoking Awareness and Intention to Quit Smoking in Smoking Female Workers: Secondary Data Analysis"

_ijerph, 2022, doi:10.3390/ijerph19052841_

Round 1
Reviewer 1 Report
Summary: This study examines correlates of the intention to quit smoking in a sample of South Korean female smokers. The authors conduct a series of statistical test to examine how occupational characteristics, smoking awareness, and the living environment are associated with the intention to quit. They construct a regression tree to determine the most important predictors of the intention to quit. The results suggest that the most important predictor was awareness of smoking cessation policies. The regression tree also identifies some “high risk” groups, in which intention to quit is particularly low. These include, e.g., women with a low awareness of smoking cessation policies, who are living alone and have a low awareness of the harms of smoking. Similarly, intention to quit is comparatively low in women with autonomy over the working hours.
Overall assessment: The study addresses an important topic. The paper is well written and for the most part easy to follow. However, several aspects of the methodology are not described clearly and are therefore difficult to understand for readers without expert knowledge in decision tree analysis. There are also a few typos in some of the numbers reported in the main text of the manuscript. I will explain these concerns in more detail below.
Major points:
- The authors use a sample of 500 female smokers from South Korea, however, only 378 of these are included in the analysis reported here. Why were the remaining 122 observations excluded from the analysis?
- The decision tree analysis needs to be explained better in the Methods section of the article, and the authors should ensure that the interpretation of their reported results is clear to the reader. I am not an expert on this method, but I have had some prior exposure to classification and regression trees, which means that the basic aim and the interpretation of Figure 1 are clear to me. However, it is not clear to me, e.g. what the gains index measures, or how the ROI should be interpreted in this context, or why it is useful in this context to assign “costs” and “benefits” to the analysis. Moreover, the authors don’t seem to interpret these results or refer to them in the discussion of the paper. My recommendations would be that the authors should:
- Restrict the decision tree analysis to those aspects that are important for the aims of the paper.
- Provide a clear and intuitive overview of this approach in the “Data analysis” section of the manuscript to ensure that readers who are unfamiliar with this methodology are able to understand the interpretation of the results.
- There are several typos in some of the numbers reported in the manuscript. For example:
- Line 175: The 137 participants should equate to 36.2% based on the table and not 63.2% as described in the text.
- Line 178: 232 participants should equate to 61.4% (Table 1) and not 16.4% as reported in the text.
- Line 181: The number “11” should be “111” here.
- In the final paragraph of the conclusions, the authors focus on increasing awareness of the harms of smoking. I was slightly surprised by this – the results seemed to suggest that awareness of smoking cessation policies is a much more important concept for the intention to quit, so I would have thought that efforts should be focused on promoting awareness of these policies?
Minor points:
- From line 197 onwards, the authors report percentages in the text and give absolute numbers in parentheses, whereas earlier in the manuscript this was done the other way around. I would recommend to keep this consistent and report absolute numbers with percentages in parentheses throughout the manuscript.
- I was surprised to see that cohabiting with a smoker seemed to be associated with a higher intention to quit than cohabiting with a non-smoker. Perhaps this difference is not statistically significant on its own, but I was wondering about potential explanations for this pattern.
- I found the discussion of working hours autonomy and job satisfaction very balanced and helpful. I also thought that another contributing factor could perhaps be that women with low autonomy and/or job satisfaction and those with high autonomy over their hours and/or high job satisfaction smoke for different reasons (stress relief vs. enjoyment, e.g.), which might be associated with different rates in the intention to quit.
Author Response
Thank you for your review.
Please see the attached file for the answer.

Reviewer 2 Report
This study identified the factors affecting the intention to quit smoking in female smokers in Korea.
Add a background sentence to the abstract.
Delete from the abstract " IBM SPSS Statistics for Windows, Version 25.0, was used for statistical analysis, and χ2-test, correlation analysis, and decision tree analysis were conducted. "
Re-write the abstract.
Please provide the ethics approval date for the ethics committee.
Add the date to the abstract also.
The discussion section must be written in a such way to reflect the content of your manuscript: start with the main conclusion; explain and interpret the reported results with references to your tables and figures and after that compare your results with similar studies. Appropriately present the limitations of the study as well as the practical utility of the reported results.
The conclusion needs improvement.
I strongly recommend the authors seek English language revision for this manuscript. I believe this would help clarify some of the expressions and sentences that are currently not appropriate or incomprehensible.
Author Response

(The authors gave the same response as above.)

Reviewer 3 Report
The authors presented a data analysis about the intention to quit smoking in female workers in Korea.
This study could represent a starting point to increase the education of people to support the intention to quit smoking. There are some limitations that have been clearly stated by the authors, this is appreciable.
The paper is well organized and written. I have some small suggestions to improve it:
1) Abstract: in my opinion it is better to avoid a numbered list of results, but it should be more fluent to explain results, as the authors have done, without indicating the numbers of finding, but only discussing them and connecting them among each other with adverbs (moreover, furthermore, in addition, etc.)
2) Line 27: I suggest to write " ...have helped to decrease the smoking rate"
3) Conclusion: I suggest to avoid repeating the results and all the percentages obtained through the analysis. It should be useful to revise this paragraph and focus on the clear conclusion that the authors want to suggest, such a take-home message highlighting the importance of this analysis.
4) Informed consent: I think that woman who have been enrolled for this study, even if is a secondary analysis, should have signed an informed consent. Can the authors verify and revise the "Informed Consent Statement" ? I think the authors could write that a written informed consent have been provided, or something similar.
Author Response

(The authors gave the same response as above.)

Round 2
Reviewer 1 Report
I would like to thank the authors for addressing my previous comments. I am satisfied with these responses, but detected two minor issues in the revised version of the manuscript, which should be addressed before publication:
- On line 36, the statement that the smoking rate decreased with age seems to be in contradiction with the later statement that the smoking rate was lower in cohorts born after the 1970s.
- In line 148/149, the second half of the sentence seems to be simply a repetition of the first half of the sentence.
Author Response
Dear Editor and Reviewers,
We wish to thank you for your thoughtful comments and valuable feedback on the manuscript originally titled, “Smoking Awareness and Intention to Quit Smoking in Smoking Female Workers: Secondary Data Analysis”
We have modified the manuscript according to your suggestions, rewriting and rephrasing sections to improve clarity, adding further information, and explaining in detail the points that were previously vague. For your convenience, we have set the round 2 revisions in the manuscript in blue. We believe that the revised version of this paper will be of interest to the readership of the International Journal of Environmental Research and Public Health.

Reviewer 2 Report
All comments are revised and read well through the paper.
Author Response

(The authors gave the same response as above.)
